# The Geochemical Drivers of Bacterial Community Diversity in the Watershed Sediments of the Heihe River (Northern China)

Federica Piergiacomo [1], Luigimaria Borruso [1], Alfonso Esposito [1,2], Stefan Zerbe [1] and Lorenzo Brusetti [1,*]

1 Faculty of Science and Technology, Free University of Bozen-Bolzano, Piazza Università 5, 39100 Bolzano, Italy; federica.piergiacomo@natec.unibz.it (F.P.); luigimaria.borruso@unibz.it (L.B.); alfonso.esposito1983@hotmail.it (A.E.); stefan.zerbe@unibz.it (S.Z.)
2 International Centre for Genetic Engineering and Biotechnology Padriciano, 99, 34149 Trieste, Italy
* Correspondence: lorenzo.brusetti@unibz.it

**Abstract:** The city of Zhangye (Gansu Region, China) has been subjected to several changes related to the development of new profitable human activities. Unfortunately, this growth has led to a general decrease in water quality due to the release of several toxic wastes and pollutants (e.g., heavy metals) into the Heihe River. In order to assess the environmental exposure and the potential threat to human health, microbiological diversity for the monitoring of water pollution by biotic and abiotic impact factors was investigated. In particular, we analysed samples collected on different sites using 454 pyrotag sequencing of the 16S ribosomal genes. Then, we focused on alpha-diversity indices to test the hypothesis that communities featuring lower diversity show higher resistance to the disturbance events. The findings report that a wide range of environmental factors such as pH, nutrients and chemicals (heavy metals (HMs)), affected microbial diversity by stimulating mutualistic relationships among bacteria. Furthermore, a selection in bacterial taxa related to the different concentrations of polluting compounds was highlighted. Supporting the hypothesis, our investigation highlights the importance of microbial communities as sentinels for ecological status diagnosis.

**Keywords:** microbial communities; alpha diversity; freshwater sediments; *Phragmites australis*; pollution

## 1. Introduction

The availability and quality of freshwater is necessary to ensure adequate human-related ecosystem services and has been the subject of scientific discussion and studies [1,2]. In particular, it is estimated that about two billion people live in areas subjected to high water stress, while four billion people experience critical conditions in the supply of quality water for at least one month a year [3]. As anthropogenic pressure advances, the number of people facing water emergencies, including those related to water pollution, will increase over time. It has been estimated that by 2050, half of the world's population will live in territories with poor freshwater quality [4]. For this reason, it is a priority not only to safeguard the quality of freshwater, but above all to intervene by monitoring situations of stress and disturbance. Thus, it is necessary to establish water quality indicators, which are traditionally divided into three categories: physical indicators (quantity of water, flow, etc.), chemical indicators (presence of metals, nitrogen, TOC, etc.), and biological indicators. The latter are mainly fish, invertebrates and a few representatives of bacteria [5]. In particular, microorganisms, algae and fauna (e.g., invertebrates and fishes) in urban-related freshwater and sediments are exposed and easily affected by any environmental factor changes, including large loads of organic matter, nutrients and pollutants discharge (pesticides, toxic materials and heavy metals, HMs). Their overall community response in terms of richness or evenness renders them suitable bioindicators of pollutant presence, providing information on ecosystem health [6–12], and studying the biodiversity of aquatic habitats could be very important for estimating, monitoring and assuring their sustainable use and the deriving measures of environmental management and restoration. It has

been suggested that geographical abiotic (pH, nutrients, water availability, oxic/anoxic conditions, temperature, chemical pollution, HMs, pesticides, antibiotics) and biotic factors (plasmids, phages, transposons) govern microbial diversity [13]. In fact, the biogeographical distribution patterns of bacterial communities could be affected by local or regional factors (or by a combination of them), which may differ by site and time [14–17]. Moreover, the growth and division of bacterial cells depends on the amount of nutrients present in the system; variations in abiotic factors induce changes in bacterial community composition and in their physicochemical habitat (e.g., stimulating processes such as sulfate reduction or nitrogen (N) fixation) [18,19]. Therefore, bacterial communities and their diversity could be suitable indicators of environmental degradation and possibly used to assess the perturbation level, even if many authors found it challenging to select the right bacterial diversity measurement index; a deeper investigation of the topic will be crucial in the development of suitable methods [6,20,21]. Indeed, in microbial ecology, it is hypothesized that a higher biodiversity improves the ecosystem resilience and resistance by varying the responses to that environmental change to maintain ecosystem functions [22] and generally, resilience evaluation involves the characterization of the quantity, structure and shift of taxonomic and/or functional redundancy into a broad group of biological arrangements such as from genes to communities [23,24].

Diversity among microorganisms has mainly been characterised from a taxonomic view at the community level, using different measurements (e.g., alpha, beta, gamma diversity). Alpha diversity relates to the diversity of one environmental area or single sample, and it is usually determined as, e.g., number of species (species richness) and as the extension of the dominance of the species (species evenness). It has been previously shown to successfully highlight perturbances in complex freshwater basins affected by different scales of pollution. For example, McClary-Gutierrez et al. [25] provided evidence for a significant decrease (Wilcoxon rank sum test, $p < 0.01$) in bacterial community diversity in impacted streams compared with unimpacted streams and in particular, a statistically significant correlation between developed land area and Shannon diversity across the sites sampled (Spearman rank sum test, $\rho = 0.50$, $p < 0.05$) was observed. Moreover, Mohiuddin et al. [26] measured significantly different average taxonomic richness and alpha diversity between samples collected from lakes and creeks ($p < 0.05$) and between lakes and stormwater outfalls ($p < 0.05$).

While this approach is useful, it is limited by sampling procedures, which seem to severely influence species richness with the necessity of standardized samples or the utilization of estimator factors to correct sampling biases (e.g., Chao1 or ACE). On the contrary, evenness seems to be relatively unaffected by under-sampling biases and is generally determined with Simpson's or Pielou's indices or rank abundance curves [23].

Moreover, considering the taxonomic and procedural limitations of traditional culturing techniques, investigations on microbiome must rely on new culture-independent techniques such as next-generation sequencing (NGS) [20,27,28]. Previous studies, including those in freshwater systems in China, have shown through NGS approaches that bacterial community structure and diversity are correlated with physicochemical factors (e.g., nutrients, temperature, pH, dissolved oxygen) and organic pollutants (e.g., PAHs, PCBs) [29–34], but the field deserves more investigation.

The goal of this work was to test the hypothesis that alpha diversity could be used as a possible proxy for freshwater sediment health monitoring. We conducted this study in the Heihe polluted watershed, within the urban area of Zhangye City (northern China) and through high-throughput pyrosequencing techniques. The Heihe River represents a long-term interdisciplinary research area, already investigated in terms of water utilization, reed stands ecology and environmental economy [35]. Hence, we focused on the correlation between bacterial community diversity and the main environmental factors (i.e., HMs and nutrient spreading) that characterize each sampling site. Our study sought to demonstrate that alpha diversity measured through NGS approaches has the potential to identify the possible drivers behind adaptive responses in the ecosystem subjected to

changing environmental conditions. By assessing these environmental changes through microbial community diversity and partitioning communities against curated databases, we provide novel insights into how alpha diversity and sequencing data can be integrated into environmental monitoring for both human and ecological health.

## 2. Materials and Methods

### 2.1. Sampling Method

Investigations were performed in Zhangye City (Gansu Province, northern China), a modern city surrounded by the Gobi Desert and the Qilianshan Mountains. The city is characterized by canal networks (both natural and artificial) running into the Heihe River and ponds, mostly covered by *Phragmites australis* (Cav.) Trin. (common reed). The sampling procedure was designed to assure the broadest geographic coverage possible, considering different types and levels of pollution. The area was divided into four different zones based on their effluents (industrial, urban, agricultural, and natural). A total of 17 sampling sites were chosen, as stated by Borruso et al. [6] in Table 1 and showed in Figure 1.

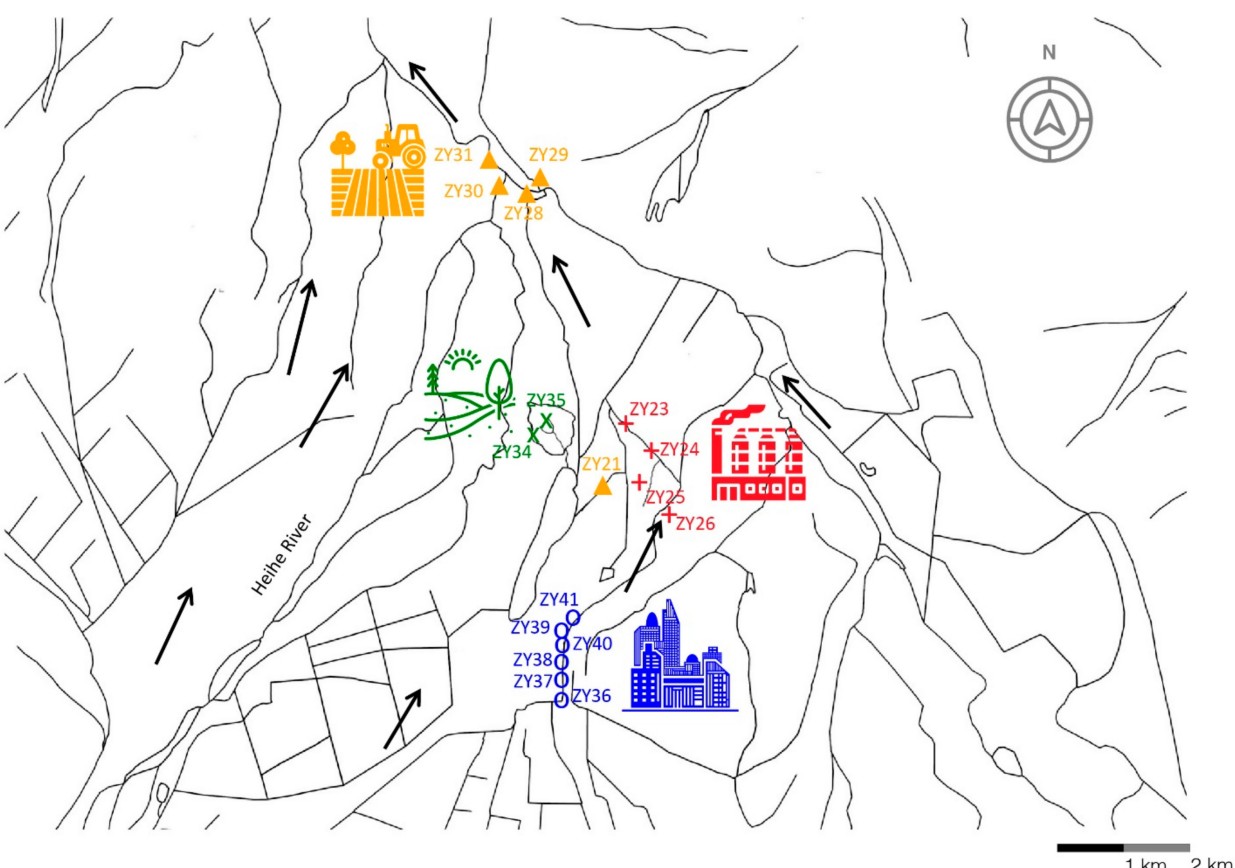

**Figure 1.** Map of the sampling sites (modified from Borruso et al. 2015). ▲: Agricultural area sampling site; X: Natural park sampling site; +: Industrial area sampling site; o: Urban area sampling site; ↑: Channel flow.

In particular, according to Table 1, the sites and areas were separated considering the anthropogenic influence. Six sampling points were examined along the main canal crossings through the main urban area (ZY36-41) with presumed organic and antibiotic pollution; four sites were near coal and metallurgy industries in a smaller secondary canal (ZY24, ZY25, ZY26) and in the main canal receiving water from the industrial area (ZY23) with probable chemical and industrial pollution. Two sites were sampled in Zhangye Natural Park (ZY34 and ZY35), where reeds were completely covering the area and where low

anthropogenic pressure was presumed. For this reason, we considered them controls. Five sites were chosen for the area where agricultural and livestock pollution were hypothesized; in particular, two were collected in an agricultural drainage canal near the Heihe River (ZY28 and ZY29), two on the banks of Heihe (ZY30 and ZY31) and one between the industrial area and the main city center (ZY21).

**Table 1.** The sampling sites, land use, GPS coordinates, times and date and the descriptions of the environments according to their pollutants.

| Sample | Land Use | Latitude (GPS) | Longitude (GPS) | Altitude (m) | Time | Date | Description of Environment |
|---|---|---|---|---|---|---|---|
| ZY21 | Agricultural area | N38 58.884 | E100 28.891 | 1454 m | 10:30 | 11 June 2011 | Drainage channel from city to wastewater plant, probable chemical pollution. Small channel. Height of reeds is about 2.5 m. Near the channel is a small pond full of reeds. |
| ZY23 | Industrial area | N38 58.989 | E100 30.001 | 1456 m | 11:30 | 11 June 2011 | Outgoing industrial channel between ZY24 and ZY22. High level of chemical pollution. Small pond surrounded by a dead forest (E, S, W) and a street with a railway (N). Reeds are about 1.5 m high. |
| ZY24 | Industrial area | N38 58.874 | E100 29.999 | 1458 m | 12:00 | 11 June 2011 | Big pond in the channel of high level of only industrial polluted water. Water was black. Dead trees. Nearby, there was big coal piles. Reeds are about 2 m high. |
| ZY25 | Industrial area | N38 58.820 | E100 29.953 | 1456 m | 12:20 | 11 June 2011 | Industrial lake with black water. High level of chemical pollution and bloom algae. Mixed vegetation of reeds and Typha. Near a factory of coal dealing. |
| ZY26 | Industrial area | N38 58.848 | E100 30.206 | 1457 m | 12:40 | 11 June 2011 | Industrial lake with high level of chemical pollution and bloom algae; the height of the reeds is about 3 m. |
| ZY28 | Agricultural area | N39 03.065 | E100 26.927 | 1435 m | 09:20 | 12 June 2011 | Shandan River, as an outgoing channel from an industrial area; Livestock, grassland and agricultural area; height of the reeds about 3 m. |
| ZY29 | Agricultural area | N39 03.065 | E100 26.927 | 1435 m | 09:40 | 12 June 2011 | 1 km far from the entrance of the Shandan River to the Heihe River. Agriculture, corn crops, some livestock; some foam in the water, not so far to a small river dam |
| ZY30 | Agricultural area | N39 03.113 | E100 26.807 | 1432 m | 10:10 | 12 June 2011 | Entrance of the Shandan River into the Heihe River; agricultural and natural area. Wild birds; lot of paw prints from livestock. |
| ZY31 | Agricultural area | N39 03.121 | E100 26.800 | 1432 m | 10:10 | 12 June 2011 | Entrance of the Shandan River into the Heihe River; agricultural and natural area. Wild birds; lot of paw prints from livestock. |
| ZY34 | Natural area | N38 59.258 | E100 27.793 | 1455 m | 12:00 | 12 June 2011 | National natural wetland park. Extensive reed stands; wild birds; not flowing water |
| ZY35 | Natural area | N38 59.314 | E100 27.704 | 1452 m | 12:00 | 12 June 2011 | National natural wetland park. Extensive reed stands; wild birds; not flowing water |
| ZY36 | Urban area | N38 55.992 | E100 28.076 | 1481 m | 09:00 | 13 June 2011 | In Zhangye city, the water source of the urban channel. The reeds are about 3 m high. Organic pollution, although recently a wastewater pipeline was built close to this channel |
| ZY37 | Urban area | N38 56.107 | E100 28.066 | 1483 m | 09:00 | 13 June 2011 | In Zhangye city, the water source of the urban channel. The reeds are about 3 m high. Organic pollution, although recently a wastewater pipeline was built close to this channel |
| ZY38 | Urban area | N38 56.146 | E100 28.085 | 1483 m | 10:20 | 13 June 2011 | In Zhangye city, the water source of the urban channel. The reeds are about 3 m high. Organic pollution, although recently a wastewater pipeline was built close to this channel |
| ZY39 | Urban area | N38 56.280 | E100 28.095 | 1483 m | 11:00 | 13 June 2011 | Presence of antibiotic vials in the water; some handcraft factories; height of reeds is about 2.5 m. |
| ZY40 | Urban area | N38 56.279 | E100 28.092 | 1483 m | 11:20 | 13 June 2011 | Presence of antibiotic vials in the water; some handcraft factories; height of reeds is about 2.5 m. |
| ZY41 | Urban area | N38 56.278 | E100 28.091 | 1483 m | 11:20 | 13 June 2011 | Presence of antibiotic vials in the water; some handcraft factories; height of reeds is about 2.5 m. |

In each sampling site, we collected one sample of rhizosphere (particles enveloping the root within 1 and 3 mm) with a careful shaking from three water dispersed roots of

*P. australis* plants distancing about 10 cm. In the same place, about 10 g of wet sediment were taken with a sterile spoon and put into sterile tubes at 4 °C, while the remaining sediment from each root was collected with the same method and placed into sterile bags at −20 °C to allow molecular and chemical analysis [6]. The tubes were rapidly brought to the nearest laboratories for the DNA extraction. Then, DNA containing tubes and sediments were shipped to Italy in a box with dry ice.

### 2.2. Chemical and Physical Analysis of Samples

Sediment samples (about 200 g) were oven-dried at 105 °C, until constant weight was attained. Acid digestion ($HNO_3$ concentrated 65% and $H_2O_2$ 30%) in a Milestone high-performance microwave oven (MLS Mega, Gemini BV Laboratory, Apeldoorn, The Netherlands) followed. Metals, including iron (Fe), aluminium (Al), manganese (Mn), copper (Cu), zinc (Zn), chromium (Cr), nickel (Ni), lead (Pb), cobalt (Co), mercury (Hg), cadmium (Cd) and arsenic (As) and total phosphorous (P) were detected by inductively coupled plasma–optical emission spectroscopy (ICP-OES, Spectro Ciros CCD, Spectro GmbH, Kleve, Germany). Total nitrogen (N) concentration was estimated through the Dumas combustion method with a TruSpec N analyser (LECO Corporation, St. Joseph, MI, USA). The $pH_{H2O}$ of the sediments was measured using an Accumet AP85 pH (Fisher Scientific Ltd., Pittsburgh, PA, USA).

### 2.3. DNA Extraction and 454 Pyrotag Sequencing

Wet sediments (about 5 g) were centrifuged at 13,000 rpm for 15 min to remove the excess water. Then, according to the user manual, PowerSoil® DNA Isolation Kit (MoBio, Arcore, Italy) was used for the extraction of the total community genomic DNA from 1 g (wet weight) of the centrifuged sediments. The quantification of DNA was performed using the NanoVue™ Plus (GE Healthcare, Little Chalfont, UK) spectrophotometer, adjusting the concentration to 10 ng/µL. The genomic DNA of each of the 17 samples were shipped to the Molecular Research LP (MR DNA™) (http://www.mrdnalab.com (accessed on 10 June 2022)). The DNA was used for the PCR amplification of the environmental 16S rRNA genes for bacteria with a primer set amplifying the V4-V6 variable regions (primers 518F 5′-CCAGCAGCYGCGGTAAN-3′ and 1046R 5′-CGACRRCCATGCANCACCT-3′). Samples were sequenced using the Roche 454 GS-FLX system, titanium chemistry, according to the protocols of that company. The pyrosequencing data were analysed with a proper analysis pipeline as suggested by the company. Sequences with length <200 bp, or with ambiguous bases and homopolymer runs exceeding 6 bp, were removed before the chimera checking. The 16S rRNA gene pyrotags were defined after the removal of singleton sequences by clustering into 97% similarity. Representatives from each OTU were classified using BLASTN against a curated GreenGenes database. All pyrotags have been submitted to the EMBL/NCBI/DDBJ Short Read Archive under study accession number PRJEB4308. To normalize the variations in read depth across samples, the data were rarefied to the minimum read depth of 1248 sequences per sample.

### 2.4. Statistical Analysis

PAST 3.26 software was used to perform cluster analysis using the UPGMA algorithm and the Bray-Curtis similarity index. R software (R Foundation for Statistical Computing, Vienna, AT; http://www.R-project.org/ (accessed on 10 June 2022)) was used to investigate the alpha diversity indices (i.e., richness, Shannon, Simpson and evenness) and the Pearson correlation among genera and environmental parameters. Permutational multivariate analysis of variance (PERMANOVA) was applied to test the possible effect of the different land uses on the bacterial communities.

## 3. Results and Discussion

According to MacDonald et al. [36] and Persaud et al. [37] classifications, metal concentration over the probable effect concentration (PEC) were identified as toxic on the

biota, as already described by Borruso et al. [6] and shown in Table 2. For example, high concentrations of Cd and Pb were found in the industrial sites (ZY23, ZY24, ZY25) and some metals like Ni and Zn were over the PEC in ZY25 and ZY24. For the urban area (ZY36 to ZY41), Hg, Ni and Cr showed higher values. Conversely, in the agricultural area (ZY21, ZY28, ZY29, ZY30, ZY31) and the natural park (ZY34, ZY35), all metals had concentrations much lower than the PEC threshold, with the exception of Ni levels in ZY28. The concentrations of Co and Mn were also below the PEC threshold for all sites. N was higher in and around the urban area, probably due to civil wastewater release (see Table 1), followed by the industrial one. The samples collected in the natural park (i.e., control) had metal concentration values below the PEC threshold.

**Table 2.** The pH$_{H20}$, metal and nutrient concentrations of the sediments in the different sampling sites with probable effect concentrations on the biota (PEC) adapted from MacDonald et al. (2000) for As, Cd, Cu, Pb, Hg, Ni and Zn and Persaud et al. (1993) for Mn and Co. Metal concentrations over the (PEC) are in bold. Sediments concentration: dry weight (a) g/kg and (b) mg/kg. Borruso et al. [1].

| Site | pH | N % | P (a) | Fe (a) | Al (a) | Mn (b) | Cu (b) | Zn (b) | Cr (b) | Ni (b) | Pb (b) | Co (b) | Hg (b) | Cd (b) | As (b) |
|---|---|---|---|---|---|---|---|---|---|---|---|---|---|---|---|
| ZY21 | 7.6 | 0.16 | 0.88 | 30.3 | 23.4 | 623 | 41 | 102.1 | 56.5 | 47.1 | 23.2 | 14.8 | 0.3 | 0.24 | 12 |
| ZY23 | 8.6 | 0.22 | 1.07 | 28.5 | 27.1 | 570 | **151** | **2023** | 50.2 | 41.8 | **720** | 15.4 | **2.22** | **74.24** | **248** |
| ZY24 | 7.8 | 0.6 | 2.37 | 29.9 | 29.9 | 601 | **111.6** | 313.7 | 81.1 | **49.9** | **201.5** | 14.3 | 0.95 | **6.39** | **184** |
| ZY25 | 8.5 | 0.27 | 1.3 | 30 | 27.1 | 697 | 85.2 | **687.5** | 50.1 | 43.2 | **139.5** | 16.8 | 0.88 | **12.9** | **296** |
| ZY26 | 7.9 | 0.15 | 0.82 | 28.7 | 25.1 | 624 | 50 | 163.1 | 85.8 | **52.3** | 52.6 | 12.8 | 0.09 | 1.22 | **80.5** |
| ZY28 | 7.9 | 0.11 | 0.64 | 22.5 | 14.4 | 476 | 23.4 | 133 | 93.4 | **52.6** | 34.1 | 9.24 | 0.19 | 2.07 | 14 |
| ZY29 | 8.5 | 0.02 | 0.57 | 19 | 13 | 508 | 12 | 34.5 | 20.8 | 14.2 | 12.2 | 7.89 | 0.01 | 0.26 | 10 |
| ZY30 | 8.2 | 0.02 | 0.55 | 20.5 | 14.2 | 536 | 13.1 | 40.5 | 26.3 | 19.8 | 13.2 | 8.5 | 0.01 | 0.14 | 8.6 |
| ZY31 | 8.2 | 0.05 | 0.67 | 26.5 | 15.5 | 565 | 14 | 41 | 30.9 | 17 | 15 | 9.3 | 0.02 | 0.23 | 12.1 |
| ZY34 | 8.2 | 0.1 | 0.57 | 27.1 | 22.3 | 663 | 29 | 56.39 | 45.2 | 35.8 | 15.3 | 13.4 | 0.02 | 0.2 | 16.9 |
| ZY35 | 8.1 | 0.12 | 0.57 | 30.4 | 25.1 | 723 | 31 | 62.6 | 44.1 | 38.9 | 16.1 | 14.6 | 0.02 | 0.21 | 15.4 |
| ZY36 | 7.7 | 0.99 | 2.47 | 32.5 | 28.4 | 462 | 64.1 | 184 | 116 | 54.9 | 28.4 | 13.6 | 0.15 | 0.29 | 9.2 |
| ZY37 | 8.1 | 0.49 | 1.66 | 29.2 | 21.5 | 527 | 50.6 | 172 | 71.1 | 46 | 29.3 | 13.3 | 2.09 | 0.31 | 11.6 |
| ZY38 | 7.8 | 0.23 | 0.8 | 35.6 | 30.4 | 768 | 36.9 | 90.1 | 65.4 | 54.1 | 22.5 | 16.8 | 0.32 | 0.12 | 12.8 |
| ZY39 | 8.3 | 0.74 | 2.21 | 36.3 | 32.8 | 614 | 71.8 | 264.2 | 95.1 | 52 | 40.4 | 15.6 | 5.95 | 0.49 | 18 |
| ZY40 | 7.9 | 0.41 | 1.48 | 25.8 | 21 | 426 | 37.1 | 127.9 | 70 | 42.7 | 23.8 | 12.3 | 5.51 | 0.26 | 8.3 |
| ZY41 | 8.6 | 0.25 | 1.01 | 26.6 | 15.6 | 506 | 31.4 | 82.6 | 57.4 | 42.7 | 18.4 | 13.3 | 1.42 | 0.13 | 6.5 |

The yield of the pyrosequencing run, after the quality check, was 51,364 pyrotags from all the samples. The total number of OTU identified as bacteria was 33,304, with an average per sample of 1850 ± 590 (OTU at 97% similarity). Alpha diversity, calculated as richness, Shannon and evenness indices, was measured. The highest values for Shannon and evenness were found in ZY25, with 489.04 and 0.663, respectively, while the lowest were in ZY39 with 90.77 and 0.533 (Table 3). Richness had an average of 582 ± 134. Sample ZY21 had the lowest richness (311), and ZY31 had the highest (819).

**Table 3.** Alpha diversity (richness, Shannon and evenness) indices.

| | Richness | Shannon | Evenness |
|---|---|---|---|
| ZY21 | 369 | 228.53 | 0.637 |
| ZY23 | 599 | 421.14 | 0.655 |
| ZY24 | 441 | 197.01 | 0.601 |
| ZY25 | 643 | 489.04 | 0.664 |
| ZY26 | 496 | 208.30 | 0.596 |
| ZY28 | 528 | 289.21 | 0.627 |
| ZY29 | 544 | 340.29 | 0.642 |
| ZY30 | 519 | 331.42 | 0.643 |
| ZY31 | 527 | 296.64 | 0.630 |
| ZY34 | 564 | 408.14 | 0.658 |
| ZY35 | 591 | 399.30 | 0.651 |
| ZY36 | 459 | 239.70 | 0.620 |
| ZY37 | 396 | 133.16 | 0.567 |
| ZY38 | 457 | 245.96 | 0.623 |
| ZY39 | 349 | 90.77 | 0.534 |
| ZY40 | 412 | 184.03 | 0.600 |
| ZY41 | 459 | 209.66 | 0.605 |

The relationships between the indices and the metals/elements/pH were analysed by Pearson correlation as shown in Table 4. Only those with correlation values $\rho > |0.5|$ and $\rho < 0.05$ were considered meaningful for the analysis. The P, Cr and Hg concentrations displayed significant negative correlations with bacterial richness (P: $p$ value of 0.031, Cr: 0.048, Hg: 0.049) but also with Shannon (P: $p$ value of 0.009, Cr: 0.018, Hg: 0.015) and evenness (P: $p$ value of 0.009, Cr: 0.019, Hg: 0.016). N concentrations were negatively and significantly correlated to the evenness and Shannon indices ($p$ value: 0.022 and 0.021 respectively) but not to richness.

**Table 4.** Pearson correlation factors between alpha diversity (richness, Shannon and evenness) indices and the environmental variables (only correlation values with R > |0.5| and $p$-value < 0.05 were considered meaningful for the analysis, n.s. = not significant).

| Environmental Variables | Diversity Indices | | | | | |
|---|---|---|---|---|---|---|
| | Richness | $p$ Value | Shannon | $p$ Value | Evenness | $p$ Value |
| N | −0.47 | n.s. | −0.55 | 0.021 | −0.54 | 0.022 |
| P | −0.52 | 0.031 | −0.60 | 0.009 | −0.60 | 0.009 |
| pH | 0.39 | n.s. | 0.29 | n.s. | 0.22 | n.s. |
| Fe | −0.34 | n.s. | −0.38 | n.s. | −0.36 | n.s. |
| Al | −0.2 | n.s. | −0.24 | n.s. | −0.24 | n.s. |
| Mn | 0.26 | n.s. | 0.25 | n.s. | 0.23 | n.s. |
| Cu | −0.10 | n.s. | −0.20 | n.s. | −0.22 | n.s. |
| Zn | 0.10 | n.s. | −0.02 | n.s. | −0.08 | n.s. |
| Cr | −0.48 | 0.048 | −0.56 | 0.018 | −0.56 | 0.019 |
| Ni | −0.35 | n.s. | −0.38 | n.s. | −0.36 | n.s. |
| Pb | 0.2 | n.s. | 0.09 | n.s. | 0.03 | n.s. |
| Co | −0.14 | n.s. | −0.15 | n.s. | −0.14 | n.s. |
| Hg | −0.48 | 0.049 | −0.57 | 0.015 | −0.57 | 0.016 |
| Cd | 0.41 | n.s. | 0.27 | n.s. | 0.18 | n.s. |
| As | 0.42 | n.s. | 0.28 | n.s. | 0.19 | n.s. |

Figure 2 shows the response of the bacterial taxonomy according to the concentrations and spreading of the nutrients and metals or to pH changes through a co-correlation plot (2% average relative abundances) generated by Pearson correlation index. Similarly, Figure 3 shows the Pearson co-correlation plot (always 2% average relative abundances) focusing on the inhibition/induction dynamics among the bacterial taxa. In this context, Table S1 reports all the detailed data for correlation factors and related $p$-values, while Table S2 sums up the main outcomes in terms of geochemical drivers, taxonomic groups and the potential indicator service.

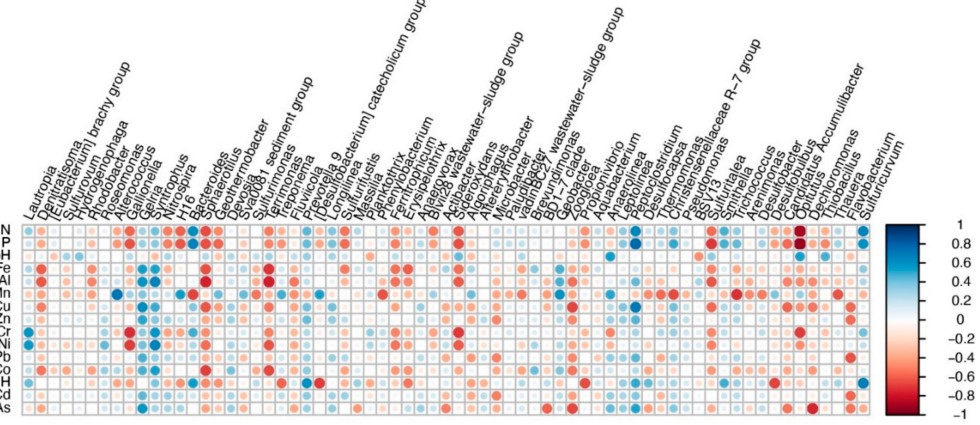

**Figure 2.** Pearson co-correlation plot (2%) showing the response of bacterial taxonomy according to the concentrations and spreading of nutrients, pH and metals.

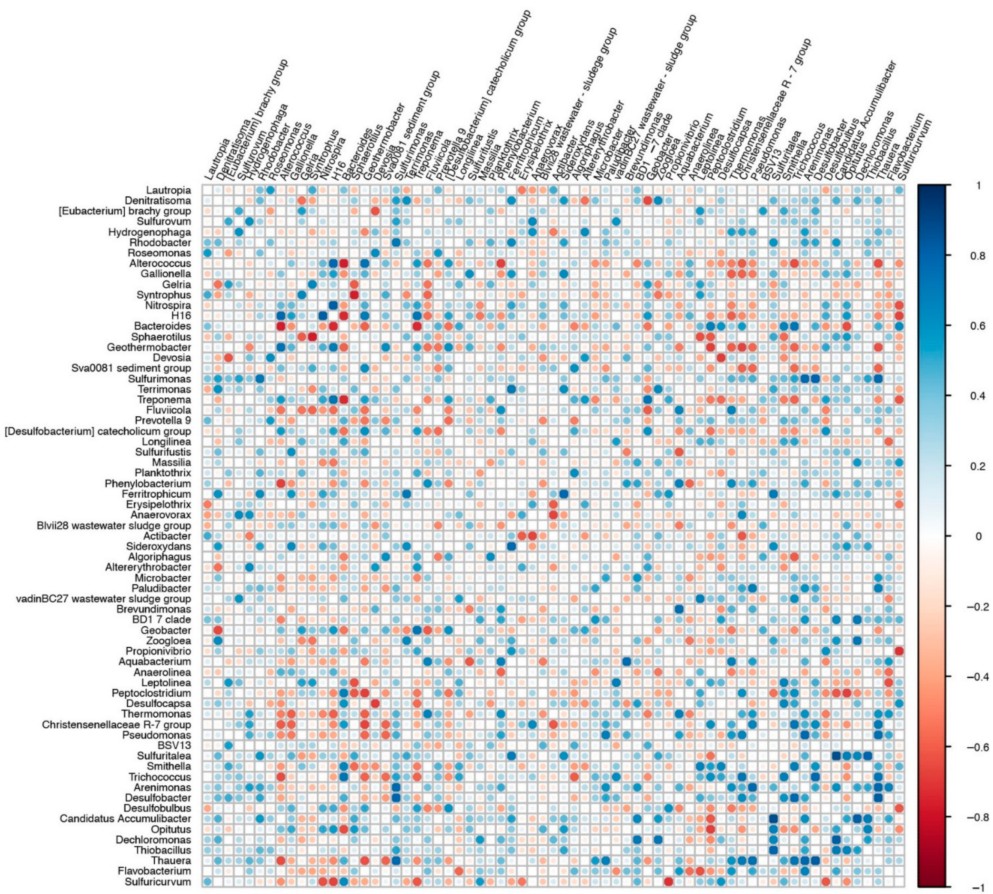

**Figure 3.** Pearson co-correlation plot (2%) showing taxa dynamics (inhibition or induction effects) induced by taxa themselves.

Here again, only significant results (R > |0.5| and *p*-value < 0.05) were considered. At the genus level, with average relative abundances more than 2%, we detected the presence of 52 main genera among sites. The results shown in the figures/tables highlight that most genera were strictly related to civil and agricultural wastewater, to gut microbiome or to human cell metabolism, and as such, to the area use/pollutant/nutrient availability and the human influence. Having demonstrated this, for the discussion, we hence focused more on taxonomic groups whose potential indicator service is related to the pollution spread. Further investigations might consider different taxonomic groups.

For example, nutrient availability (N and P) seemed to positively affect the genera *Bacteroides* and *Sulfuricurvum*, while both negatively influenced the genera *Ralstonia eutropha* H16 (also named as H16 or *Cupriavidus necator*) and *Opitutus*. Additional genera negatively influenced by N and P were *Sideroxydans* and *Sulfuritalea*, which are characterized by a metabolism involved in Fe and S transformation. Moreover, pH showed a weakly positive but significant effect on the genera *Anaerolinea* and *Opitutus*. The genus *Bacteroides* was affected positively by Hg and negatively by Mn. The abundance of the genus *Gelria* was positively influenced by metal concentration (Al, Fe, Zn, Cu and As) and also co-correlated with the genus *Syntrophus*. On the contrary, the genus *Zooglea* was negatively affected by Al, Cu, Co and As, and the genus *Flavobacterium* was negatively correlated with Cu, Zn, Pb, Cd and As. For the 52 main genera found, the mean abundance and the standard deviation per land use were also calculated (Table S3). The results confirm that there was a selection of microbial communities in line with the use of land and the related affecting pollutants and variables/elements. More specifically, the genera *Bacteroides* and *Sulfuricurvum* were positively correlated with N and were more abundant in the urban and industrial areas (0.25 and 0.24, respectively, as average of frequencies of reads for *Bacteroides* and 8.79 and

0.35 for *Sulfuricurvum*), where the nitrogen concentration turned out to be higher. In contrast, *Ralstonia eutropha H16*, strongly affected by N, was predominant (average of frequency of reads = 0.67) in the natural park (where there were low levels of N). Similarly, *Gelria*, positively influenced by Zn, showed much higher abundance in the industrial area (Zn concentration over the PEC − average of frequency of reads = 0.51).

　　Moreover, the PERMANOVA test confirmed significant differences (*p*-value < 0.001) in the distributions of the bacterial species in relation to the land use (Table 5). Cluster analysis (UPGMA on Bray-Curtis distance) of the ecological matrix derived by the 16S rRNA gene pyrosequencing data separated the communities into five groups (Figure 3). The first included ZY23, ZY24 and ZY25 sites (industrial area); the second, ZY28, ZY 29, ZY30, ZY31 (agricultural area); the third, only ZY34 and ZY35 (natural park); the fourth, ZY36 and ZY37 (urban area); and the last, ZY39, ZY40 and ZY41 sites (urban area). Finally, ZY21 (agricultural), ZY26 (industrial) and ZY38 (urban) were scattered into the clustering.

**Table 5.** The permutational multivariate analysis of variance results (PERMANOVA) (with 999 of permutations).

| Permanova | | | | | |
|---|---|---|---|---|---|
| | Df | Sum Of Sqs | R2 | F | Pr (>F) |
| Land_use | 3 | 1.8013 | 0.2981 | 1.8404 | 0.001 |
| Residual | 13 | 4.2413 | 0.7019 | | |
| Total | 16 | 6.0426 | 1.0000 | | |

　　Considering the geographic distribution of the urban, industrial, agricultural, and natural areas around Zhangye City, Borruso et al. [6] demonstrated how bacterial communities could be used as effective diagnostic tools in complex environments like cities, with the ability to discriminate each kind of environmental pollution according to land use. In this study, we evaluated the effects of specific environmental parameters (chemical elements/metal concentration) on microbiome diversity and on dynamic responses. We wished to test the effectiveness of microbial communities reflecting the environmental pollution by metal exposure. The hypothesis is that richer microbial communities may better resist long-term disturbances, reflected in the pollution in sediments.

　　According to MacDonald et al. [36] and Persaud et al. [37], we found that toxic concentrations of HMs in sediments were mainly due to different discharge sources into the channels, as well as to industrial by-products from coal combustion for energy production, as well as residuals from metallurgy industry and from chemical plants. Higher levels of chromium were caused by vehicle exhaust particles and the combustion of civil wastes [38]. N was directly released into the main urban channel through civil wastewater [39], but it was also present in the agricultural area due to fertilization.

　　P, Cr and Hg concentrations had a negative correlation with bacterial richness (Table 4). This finding is consistent with what was found in a previous study by Borruso et al. [40], where P was identified as drivers of the rhizobacterial community distribution. We confirm also what is showed by other works that demonstrated the key role of HMs (such as Cr and Hg) on the functional diversity of bacterial communities in freshwater sediments [41], as well as in other ecosystems such as soils [42] or wastewater treatment plants [43]. Other studies instead showed that the diversity is not always negatively correlated with HMs but could increase at low doses of HMs [44], due to the possible enrichment of ecological niches favouring the incidence of different gene functionalities. Our results are somewhat consistent with the intermediate disturbance hypothesis. Surprisingly, we did not observe the significant correlations between pH variation and microbial shift (Table 4) that were shown elsewhere [18,45].

　　The co-occurrence analysis showed the presence of 52 main bacterial genera mostly associated to civil and agricultural wastewater, gut microbiome or human metabolism. For example, nutrient availability (N and P) seemed to positively affect the genera *Bacteroides*

(*p*-value = 0.008) and *Sulfuricurvum* (*p*-value = 0.005), while both negatively influenced the genus *Ralstonia eutropha* (*p*-value = 0.014). *R. eutropha* strain H16 is known to be a hydrogen-oxidizing *Betaproteobacterium* commonly found in freshwater and soils. It can synthesize a wide range of metabolites and bioplastics (polyhydroxyalkanoates). This bacterium features both heterotrophic and lithoautotrophic metabolisms [46,47]. Additional genera that were negatively influenced were *Sideroxydans* (*p*-value = 0.009 for N, and 0.006 for P) and *Sulfuritalea* (*p*-value = 0.020 for N, and 0.002 for P), characterized by a metabolism involved in Fe and S transformation. Furthermore, pH showed a weak influence and positive significant effects on the genera *Anaerolinea* and *Opitutus* (*p*-value = 0.023 and 0.026 respectively). The first is usually found in natural anaerobic environments, and one of the core microbial populations in anaerobic digesters [48,49], meanwhile, *Opitutus* encompasses anaerobic fermentative species linked to nitrate reduction as a typical *P. australis* rhizobacterial genus [40,50]. The genus *Bacteroides* was also positively affected by Hg (*p*-value = 0.023) but negatively by Mn (*p*-value = 0.003). Wu et al. found that a combination of factors (HMs, physicochemical properties) induced the spreading and persistence of antibiotic-resistant genes (ARG's) among *Bacteroides* species [51]. Interestingly, the abundance of the genus *Gelria*, which has a role in syntrophic acetate oxidation [52], seemed to be positively influenced by metal concentration (Al—*p*-value = 0.014, Fe—0.028, Zn—0.034, Cu—0.009, and As—0.015). *Gelria* is a thermophilic, anaerobic, obligately syntrophic, glutamate-degrading genus, usually found in granular sludge within anaerobic sludge bed reactors [53]. The genus *Syntrophus* (positively influenced by Al—*p*-value = 0.004, Fe—0.030, Cr—0.016, Ni—0.004, and Co—0.027) is responsible for methane production and was found to be correlated to *Gelria* for treating the mixed-LCFA (long-chain fatty acids) at low temperature in municipal wastewater plants [54]. *Syntrophus*-associated sequences have been identified in anoxic sediments and sewage sludge [48]. Its presence could indicate potential environmental exposure to several xenobiotic compounds released by industrial activity. Indeed, trace amounts of HMs may stimulate the growth and activity of methanogens, and their high levels have toxic effects [43]. On the contrary, the genus *Zooglea* was negatively affected by Al (*p*-value = 0.02), Cu (*p*-value = 0.003), Co (*p*-value = 0.03) and As (*p*-value = 0.004). *Zooglea* are betaproteobacteria usually detected in both fresh and polluted waters/sediments [48]. Their cells produce finger-like projections or outgrowths in sludge flocs [55]. The genus *Flavobacterium* was negatively correlated with Cu (*p*-value = 0.01), Zn (*p*-value = 0.02), Pb (*p*-value = 0.007), Cd (*p*-value = 0.01) and As (*p*-value = 0.02). *Flavobacteria* are involved in ammonia, nitrite, and metal removal [56] and are often isolated in freshwater environments. Some of them are known to cause disease in freshwater fishes [48,57]. Most of the genera with a sulphur-oxidative metabolism (e.g., *Sulfurimonas*, *Sulfuritalea*) suffer from the spread of Hg, Mn and Cu. On the other hand, sulphate-reducing bacteria belonging to genera *Desulfobacter* play an important role in the transformation of metals such as Cu, Fe, Mn, Pb, Sb, and Zn into oxidizable fractions [58].

The responses and interactions identified through the chemical elements were confirmed by what has been observed by taxa dynamics. For instance, the genera *Gelria* and *Syntrophus* were positively co-correlated with *Zooglea*, whereas *Syntrophus* was negatively correlated. Due to the presence of high concentration of N and P, *Bacteroides* and *R. eutropha* H16 were negatively correlated. Considering the sulphur metabolism, the sulphur- and hydrogen-oxidizer *Sulfurimonas* positively influenced the sulphate-reducer *Desulfobacter* and vice versa. Here, cross-feeding microbial interactions could stimulate different genera of microorganisms to help them in adapting to environmental changes [59]. Separately, changes in microbial community dynamics owing to HM pollution (e.g., such as sensitive species replacement) largely depend on the community richness [60] and evenness could also be affected by community shifts [61].

Cluster analysis highlighted the separation of the 16S rRNA gene diversity according to the land use. Among the clusters, one group contained microbial communities from sites characterized by high levels of Cu, Zn, Ni, Pb, Hg, Cd and As and the presence of industrial effluents. The second group was composed of microbial communities strongly influenced

by agricultural effluents. The third one included samples taken from a reed stand natural area. The fourth group consisted of samples from an area with high levels of Cr, Ni and Hg and contamination from city effluents. The last cluster included sites with high levels of Ni and Hg and contamination from city effluents. As already proposed by Borruso et al. [6], ZY38 showed a different community due to the generally lower concentrations of metals and nutrients, whereas the divergence of ZY26 was probably due to a dramatic algal bloom observed during the sampling. ZY21 seemed to be affected by the urban and agricultural discharges, resulting in a borderline sample between the two areas.

## 4. Conclusions

In the Zhangye area, we investigated the diversity and the distribution of bacterial communities in 17 interconnected freshwater habitats with different land uses. We then examined the influence of environmental factors on bacterial communities, with the aim of testing alpha-diversity measurement as a tool for the efficient biomonitoring of pollution. Our results indicate that lower alpha diversity index values of microbial communities correlate with higher rates of trace pollutants and a wide range of environmental factors such as nutrients and chemicals (HMs), stimulating mutual relationships among bacteria as the co-correlation analyses show (Figures 2 and 3 and Tables 3 and 4). These findings suggest that bacteria are sensitive to changes in biotic/abiotic factors. Indeed, the distribution of the bacterial species in relation to the land use and thus to pollutant/nutrient/metal presence was confirmed by PERMANOVA with significant difference ($p < 0.001$—Table 5) and habitat-specific bacterial species were identified.

Moreover, the cluster analysis (Figure 4) highlighted similarities in the ongoing bacterial taxon selection processes, probably due to the different concentrations of polluting compounds in the different areas. We demonstrated that alpha diversity measurement and the metagenomic approach could be valid tools for assessing water/sediment quality for human and environmental safety.

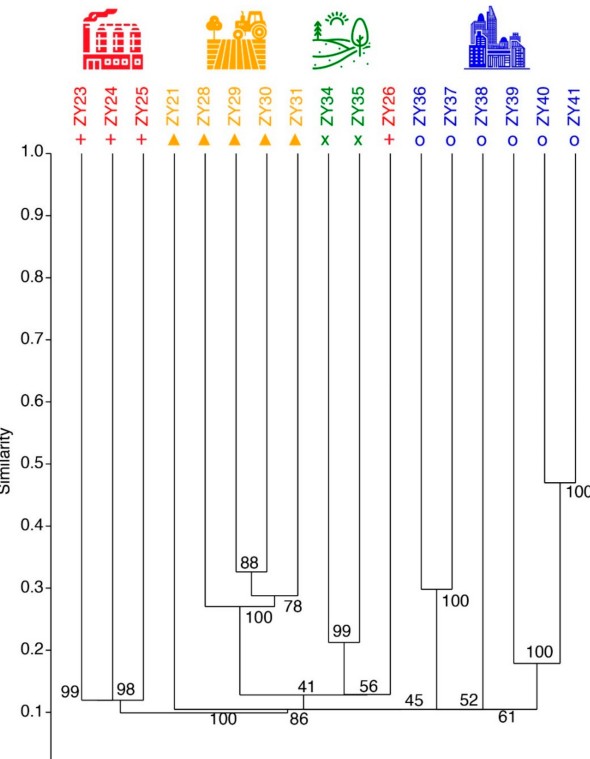

**Figure 4.** Cluster analysis (UPGMA Bray-Curtis) of the 16S rRNA gene pyrosequencing of bacterial communities. Red crosses indicate samples from the industrial zone, orange triangles those from the agricultural one, green cosses the natural zones, while blue circles the urban ones.

**Supplementary Materials:** The following supporting information can be downloaded at: https://www.mdpi.com/article/10.3390/w14121948/s1, Table S1: Pearson co-correlation plot (2%) showing the bacterial taxonomy according to the concentrations and spreading of nutrients/pH/metals, as well the taxa dynamics (inhibition or induction effects) induced by the taxa themselves. Table S2: Summary of the main outcomes Table S3: Mean relative abundance (frequencies of reads) and standard deviation per land use measured for the main 52 genera detected.

**Author Contributions:** Conceptualization, S.Z. and L.B. (Lorenzo Brusetti); methodology, L.B. (Luigimaria Borruso) and F.P.; software, L.B. (Luigimaria Borruso) and A.E.; validation, F.P., L.B. (Luigimaria Borruso) and L.B. (Lorenzo Brusetti); formal analysis, L.B. (Lorenzo Brusetti); investigation, F.P. and L.B. (Luigimaria Borruso); resources, L.B. (Lorenzo Brusetti) and S.Z.; data curation, F.P. and L.B. (Luigimaria Borruso); writing—original draft preparation, F.P.; writing—review and editing, L.B. (Luigimaria Borruso), L.B. (Lorenzo Brusetti), A.E., F.P. and S.Z.; visualization, F.P. and L.B. (Luigimaria Borruso); supervision, L.B. (Lorenzo Brusetti); project administration, S.Z.; funding acquisition, S.Z. All authors have read and agreed to the published version of the manuscript.

**Funding:** This study was funded by the Erich-Ritter Foundation and the Herzog-Sellenberg Foundation (German Stifterverband) within the project Sustainable Water Management and Wetland Restoration of Settlements of Continental-Arid Central Asia (SuWaRest; no. T122/20076/2010kg). This work was also supported by the Open Access Publishing Fund of the Free University of Bozen-Bolzano.

**Institutional Review Board Statement:** Not applicable.

**Informed Consent Statement:** Not applicable.

**Data Availability Statement:** All data generated or analysed during this study are included in this published article and in the supplementary information available at www.mdpi.com/xxx/s1 website.

**Acknowledgments:** We thank He Ping and Xu Jie of the Chinese Research Academy of Environmental Sciences (CRAES) in Beijing, and the Zhangye Environment Monitoring Center for the logistic support in sampling.

**Conflicts of Interest:** The authors declare no conflict of interest.

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
