# Peer review of "The Geochemical Drivers of Bacterial Community Diversity in the Watershed Sediments of the Heihe River (Northern China)"

_water, doi:10.3390/w14121948_

Round 1

Reviewer 1 Report

To assess the environmental exposure and potential threat to human health, the article divided the city of Zhangye (Gansu Region, China) into four zones (industrial, urban, agricultural, and natural). Molecular and chemical analyses were also performed on samples from 17 sampling sites. Finally, Alpha diversity, calculated as Richness, Shannon, and Evenness indices, was measured. The calculation results of this article can prove that communities featuring higher diversity show higher resistance to the disturbance events.

Specific comments: 

1. Three measurement methods are mentioned in lines 44-45 of the text. Why is Alpha Diversity the only choice? What is the advantage of this method over other methods? It is recommended to give the reasons for the selection in the text.

2. The introduction section is an important part of the reader's understanding of the article, especially the background section of the presentation. The background information section of this article is unclear and is recommended to be rewritten.

3. More updated references are needed in the text. Try searching with some keywords。

4. The novelty of this work and the implications for the general field of research should be highlighted in the introductory section.

5. Figure 1 is too simple to draw, and it is recommended to modify it to be more beautiful.

6. Please note the formatting issues. For example, the formatting of section 2.2 headings is different from the formatting of other secondary titles.

7. More than these elements expressed in lines 167-168 of the text are negatively correlated in Figure 2. Check the data in Table 4 for completeness and explain the differences and connections between Figure 2 and Table 4.

8. Authors should briefly state the major results and how this work contributes to the overall field of study in the conclusion part.

9. In addition, how are your experiments conducted? Please provide the necessary explanation on how to obtain experimental data in Italy.

Author Response

  1. Three measurement methods are mentioned in lines 44-45 of the text. Why is Alpha Diversity the only choice? What is the advantage of this method over other methods? It is recommended to give the reasons for the selection in the text.

Alpha diversity has been demonstrated to be more able to detect perturbances on the microbial communities than other indices. We included a documented introduction to alpha diversity index with a few examples of application in similar environments (lines 73-87, track change version).

  1. The introduction section is an important part of the reader's understanding of the article, especially the background section of the presentation. The background information section of this article is unclear and is recommended to be rewritten.

We have deeply revised the introduction, including more background of the topic.

  1. More updated references are needed in the text. Try searching with some keywords

We have included more recent references. Actually, those included in the original manuscript were those we considered pioneer for this topic.

  1. The novelty of this work and the implications for the general field of research should be highlighted in the introductory section.

We highlighted the novelty at the end of the introduction (lines 124-127).

  1. Figure 1 is too simple to draw, and it is recommended to modify it to be more beautiful.

We redesigned figure 1 to make it clearer.

  1. Please note the formatting issues. For example, the formatting of section 2.2 headings is different from the formatting of other secondary titles.

We fixed the formatting.

  1. More than these elements expressed in lines 167-168 of the text are negatively correlated in Figure 2. Check the data in Table 4 for completeness and explain the differences and connections between Figure 2 and Table 4.

We completely revised the section including the check of the data from the table 4. Figure 2 has been deleted since its information have been included in the revised table 4.

  1. Authors should briefly state the major results and how this work contributes to the overall field of study in the conclusion part.

The conclusions have been completely rewritten considering the criticism of the reviewer.

  1. In addition, how are your experiments conducted? Please provide the necessary explanation on how to obtain experimental data in Italy.

Sampling methods in China (section 2.1) have been completely rewritten to provide much more information to allow replicability elsewhere.

Reviewer 2 Report

I have read the manuscript entitled „Geochemical drivers of bacterial community diversity in watershed sediments of Heihe River (Northern China)”, but the text needs improvement. It is not written clearly, there is a lot of mess and chaos that needs to be brought under control.

Table 1 is quite important  and brings many valuable inormation. Meanwhile, there is no reference , not mentioning some description, to table 1 in the text.

I don’t understand Figure 2 – what do the authors want to show here?

Figure 3 – I cannot see Ralstonia (Cupriavidus necator) which you describe  in the text in relation to Fig.3

Figure 3 needs also deeper explanation what is visible here. The quality of Figure 3 is not satisfactory (too small font).

The purpose of the work and the results obtained needs to be very clearly described. Conclusions needs also rearrangement and more clarity.

Author Response

Table 1 is quite important  and brings many valuable inormation. Meanwhile, there is no reference , not mentioning some description, to table 1 in the text.

Sampling methods in China (section 2.1) have been completely rewritten, including more information from Table 1 and description of sites and land use.

I don’t understand Figure 2 – what do the authors want to show here?

We eliminated figure 2 because we revised table 4 to include also information provided by figure 2.

Figure 3 – I cannot see Ralstonia (Cupriavidus necator) which you describe  in the text in relation to Fig.3

In figure 3, Ralstonia is indicated with H16. We highlighted it in the text (line 219).

Figure 3 needs also deeper explanation what is visible here. The quality of Figure 3 is not satisfactory (too small font).

We revised the figure 3 (now split in 2 and 3) to increase the readability. The section of results derived from Figure 3 and Tables S1 and S2 has been revised especially when explain what these four datasets are showing to the reader.

The purpose of the work and the results obtained needs to be very clearly described. Conclusions needs also rearrangement and more clarity.

We revised the aims, increasing their clarity, and, on the other hand, we revised all the results, discussion and conclusions, to meet the reviewer’s requests.

Round 2

Reviewer 1 Report

The authors have addressed all the issues in the comments, and I suggest that this paper be accepted without further revision.

Reviewer 2 Report

The authors have referenced to all my comments. The manuscript has been improved according to the suggestions.